



# A coupled human-natural system to assess the operational value of weather and climate services for irrigated agriculture

Yu Li[1], Matteo Giuliani[2], and Andrea Castelletti[2,1]

[1]Institute of Environmental Engineering, ETH Zurich, Wolfgang-Pauli-Str. 15, CH-8093 Zurich, Switzerland
[2]Department of Electronics, Information and Bioengineering, Politecnico di Milano, Piazza L. da Vinci, 32, I-20133 Milano, Italy

**Abstract.** Recent advances in weather and climate services (WCSs) are showing increasing forecast skills over seasonal and longer time scales, potentially providing valuable support in informing decisions in a variety of economic sectors. Quantifying this value, however, might not be straightforward as better forecast quality does not necessarily imply better decisions by the end-users, especially when forecasts do not reach their final users, when the provider is not trusted, or when forecasts are not

appropriately understood. In this study, we contribute an assessment framework to evaluate the operational value of WCSs for irrigated agriculture by complementing traditional forecast quality assessments with a Coupled Human-Natural System behavioral model which reproduces farmers' decisions. This allows a more critical assessment of the forecast value mediated by the end-users' perspective, including farmers' risk attitudes and behavioral factors. Our results show that the quality of state-of-the-art WCSs is still limited in predicting the weather and the crop yield of the incoming agricultural season, with ECMWF

annual products simulated by the IFS/HOPE model resulting the most skillful product. However, we also show that the accuracy of estimating crop yield and the probability of making optimal decisions are not necessarily linearly correlated, with the overall assessment procedure that is strongly impacted by the behavioral attitudes of the farmers, which can produce rank reversals in the quantification of the WCSs operational value depending on the different perceptions of risk and uncertainty.

## 1   Introduction

Weather and climate services (WCSs), defined as information on past, present, and future weather and climate useful to assist decision-making (GFCS, 2014), can provide valuable aid to a variety of economic sectors, ranging from hydropower production (e.g., Garcia-Morales and Dubus, 2007), drought management (e.g., Mwangi et al., 2014), flood protection (e.g., Cloke et al., 2017), disease spread control (e.g., Thomson et al., 2006). These services are particularly important in agriculture (Hammer et al., 2001), where weather-sensitive decisions, such as crop choices or irrigation scheduling (e.g., Dutra et al., 2013;

Winsemius et al., 2014; Wetterhall et al., 2014), are frequently to be taken. Here, WCSs are expected to be even more helpful over the next years, when extreme weather conditions will be more frequent and intense (Dai, 2011).

Over past decades, WCSs have undergone a broad development in many parts of the world (Cloke and Pappenberger, 2009). The existence of slow, and hence predictable, variations in sea surface temperature, sea ice, soil moisture, and snow cover, which interact with the atmosphere and impact on the global climate, can be used for extending predictability at the seasonal





time scale (Palmer and Hagedorn, 2006). Despite some limitations still exist (e.g., Palmer et al., 2005; Lee et al., 2011), the recent increase of model resolutions (e.g., Prodhomme et al., 2016a), the improvement of the initialization procedures (e.g., Prodhomme et al., 2016b), and the more accurate representation of some physical processes (e.g., Hourdin et al., 2013) considerably advanced the accuracy of WCSs, with current state-of-the-art products showing good forecast skills even over

seasonal and longer time scales (Doblas-Reyes et al., 2013).

A causal link between better forecast quality and higher operational value is however not necessarily straightforward (e.g., Ritchie et al., 2004; Ramos et al., 2013), especially when forecasts do not reach their final users, when the provider is not trusted, or when forecasts are not appropriately understood (e.g., Ramos et al., 2010; Frick and Hegg, 2011). In other words, while quantifying forecast quality is a necessary step in the assessment of WCSs, other indicators should be considered for

capturing the stakeholders' judgment on the value of the forecast products, i.e. their operational value, particularly when this evaluation differs from the opinion of forecasters (Hartmann et al., 2002). Yet, most assessments reported in the literature focus solely on forecast quality, defined as the similarity between the forecast estimates and the actual observations of weather or hydrological variables based on some statistically formulated performance metrics (Murphy, 1993).

Recent attempts of assessing the operational value of WCSs tend to apply long-term forecasts for feeding simulation models

in order to predict decision-relevant information, such as soil water availability for irrigation scheduling (Wang and Cai, 2009; Calanca et al., 2011) or crop production for cropping pattern decision (e.g., Hansen, 2004; Baigorria et al., 2008). The use of process-based simulation models contributes a better understanding of WCSs by stakeholders and users as it allows transforming weather forecasts (e.g., precipitation and temperature) into decision-relevant information (e.g., crop yield) through a transparent, objective, and reproducible procedure. For example, although farmers can be able to quantify the risks associated

to predictions of a dry season, they would benefit much more from anticipated information on the crop yield and the associated risk of crop failure (e.g., Challinor et al., 2005). In addition, the relationship between weather and decision-relevant variables is often nonlinear and an error in the weather forecast will not be linearly propagated into an error of the same magnitude in the crop yield prediction. The quality of forecast products evaluated on weather variables can differ from the evaluation performed on crop yield: two forecast products characterized by different levels of accuracy in predicting weather variables

can provide similar predictions of crop productivity; vice versa, two products having similar skills in predicting temperature and precipitation can attain different performance in predicting crop yield. Quantifying the value in terms of forecast accuracy in predicting decision-relevant information is therefore crucial for improving stakeholders' trust in WCSs.

Although the model-based prediction is surely a step further towards the end-users' perspective, the accuracy of the predicted decision-relevant information still represents an alternative metric of forecast quality. In this case, the prediction of weather

and climate variables per se are mapped onto the derivatives of climate forecasts (e.g., predicted yield, expected profitability), assumed as the determinants of stakeholders' decisions. However, high quality forecasts may still be unused by stakeholders (e.g., Rayner et al., 2005; Coulibaly et al., 2015). For example, an attempt of increasing the forecast accuracy for providing more early warnings often implies the risk of increasing the number of false alarms, ultimately discouraging the use of WCSs in operational context due to different perceptions of risk and uncertainty (Demeritt et al., 2007). In addition, many studies have shown how stakeholders' adoption of weather forecast bears upon their social context (e.g., Hansen, 2002; Suarez and Patt,



2004; Crane et al., 2010). Such evidence motivates exploring how users' behavioral factors influence the uptake and the use of WCSs, and suggests the need of quantifying the operational value of WCSs as the improvement in the system performance obtained by informing stakeholders' decisions with WCSs (e.g., Zhu et al., 2002; Mylne, 2002; Giuliani et al., 2015; Denaro

et al., 2017).

In this work, we propose a new framework for assessing the operational value of WCSs, which puts human in the loop by integrating traditional forecast quality assessments with a behavioral model reproducing farmers' decisions. The proposed framework relies on a three-stage procedure, which starts by investigating the quality of post-processed forecast products. These forecasts are then used as input to an integrated model representing a Coupled Human-Natural Systems (CHNSs, see

Liu et al. (2007)). This includes process-based models of the physical environment to predict decision-relevant information, coupled with decision models, which describe the farmers' decision-making process. Given the predicted climate forcing as inputs, the integrated CHNS model simulates the productions of different crops, among which each farmer selects the crop to cultivate by maximizing the expected net profit at the end of the agricultural season (Giuliani et al., 2016). This combination of process-based and decision models contributes a comprehensive and complete framework for assessing WCSs and allows the

evaluation of both the forecast quality and operational value. In addition, the decision model includes heterogeneous behavioral factors, specifically diverse levels of farmers' risk aversion (or degree of trust) with respect to forecast uncertainty, which allow the exploration the sensitivity of the overall assessment of WCSs with respect to variability of stakeholders' behaviors.

We demonstrate the potential of our approach by developing an application in the Muzza agricultural district, in northern Italy. The district is organized in 39 irrigation units, each including a number of farms receiving a continuous water supply

through an extensive irrigation network. A set of state-of-the-art long-range climate forecast products are collected from the European Center for Medium-Range Weather Forecasts (ECMWF) Stream 2 project, National Centers for Environmental Prediction (NCEP) and Canadian Seasonal to Inter-annual Prediction System (CanSIPS). The forecast horizon ranges from 7 months to 10 years. Post-processing is then used to address the mismatch of temporal and spatial resolution between the simulation models and the raw forecast products, as well as to resolve the systematic bias and uncertainty in the ensemble

forecasts. Finally, by simulating the combined process-based and decision models over the period 2001-2005, with 2003 and 2005 being extreme drought years, we perform the proposed three-stage assessment of forecast quality and operational value of WCSs. First, we assess the traditional forecast quality by comparing forecast meteorological variables against observed data. Then, we measure, via model simulations, the prediction accuracy of crop yield as an intermediate assessment of decision-relevant information for supporting farmers in improving their practices. Finally, we quantify the operational value in terms of

payoff (or opportunity cost) of using WCSs for informing the selection of the cropping pattern. This value is contrasted with the upper-bound of the system performance obtained using 'perfect forecasts' as well as a baseline situation where farmers use few simple empirical forecast models, including climatology or past observations. In addition, our decision models allow exploring alternative uses of WCSs, which depend on the personal behavioral attitude of the farmers and on their level of trust in the forecast products. In particular, we explore three different levels of farmers' risk aversion, namely risk averse, risk neutral, or risk prone, which create a spectrum of possible behavioral attitudes (e.g., Mosley and Verschoor, 2005)





The paper is organized as follows: in the next section we describe the study area, while section 3 provides details about the methodology, including the data preparation and the modeling framework. Results and discussion are then reported in sections 5-6. Finally, conclusion and directions for future research are presented in section 7.

## 2 Study site

In this work, the assessment of WCSs is conducted on a pilot CHNS, namely the Muzza irrigation district located southeast to the city of Milan (see Figure 1). The selected district is one of the largest agricultural area in the region with an arable land of around 700 km$^2$. Maize (ca. 50% of the surface) and temporary grasslands (ca. 25% of the surface) are currently the major cultivated crops, with minor crops including rice, soybean, wheat, tomato, and barley. Irrigation is provided through an extensive irrigation network (more than 4,000 km in total length) served by the Adda River and feeding 39 irrigation units.

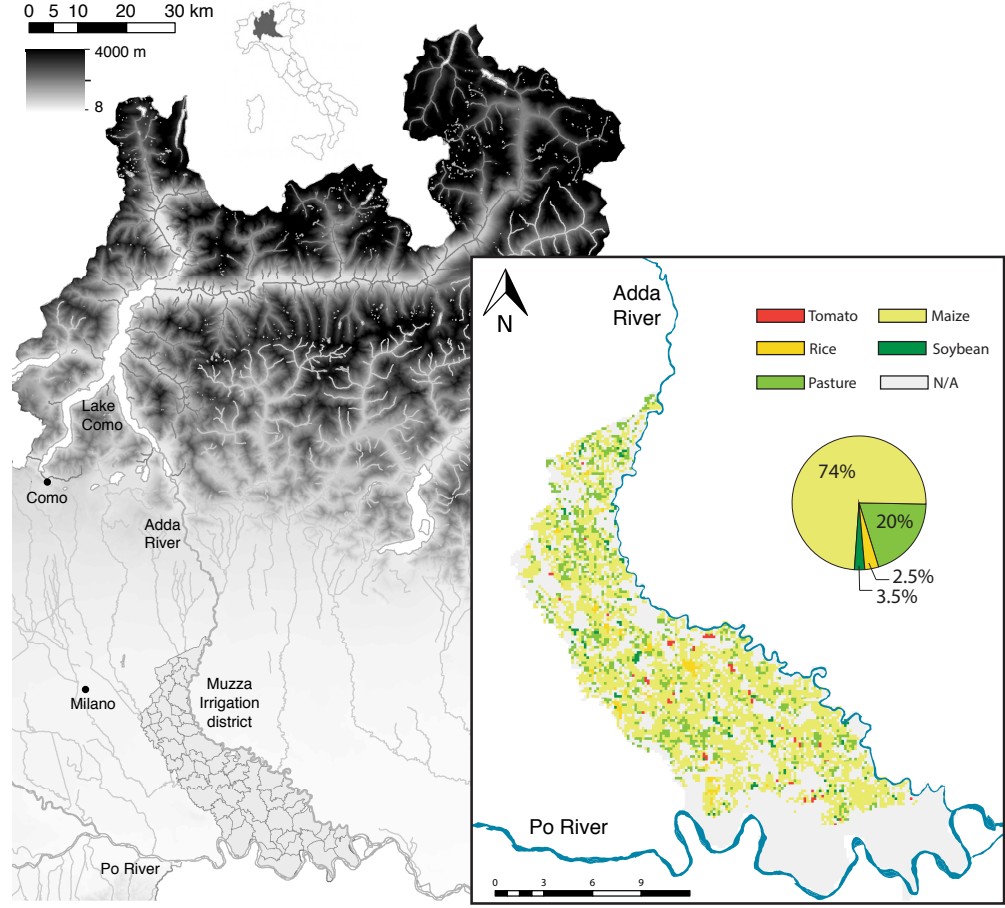

**Figure 1.** The Muzza district with the cropping pattern observed in 2004.





Historically, water availability has not been a major limiting factor to the economic development of this area. Rather, farmers were used to operate the canals' network to contribute in flood protection. Yet, in recent decades, climate change has been showing its potential negative impact in a number of situations (García-Herrera et al., 2010). For example, two severe droughts

in 2003 and in 2005 generated acute crop failures and exacerbated the conflicts between agriculture and the other sectors (Anghileri et al., 2013). These critical events are predicted to become more and more frequent over the next years (Lehner et al., 2006), representing a major challenge for the sustainability of the agricultural practices in this region.

In this context, the use of WCSs offers a promising option for supporting agricultural activities as the improved forecast skill over medium to long lead times provides valuable information about the future agricultural season prior to the sowing date.

Such information is key for better informing cropping pattern decisions to select the best crops with respect to the farmers objectives (e.g., the one characterized by the highest expected profit). Moreover, most WCSs are freely available online and thus represent a cost-effective solution to improve the resilience of agricultural systems without introducing infrastructural changes, such as modifying or expanding the irrigation canals' network.

## 3 Methodology

The overall workflow of our assessment framework is composed by three main steps, as detailed in Figure 2: ($i$) forecast quality assessment of post-processed WCSs using retrospective forecast (i.e., hindcast) products; ($ii$) extension of the forecast quality analysis via model-based prediction of decision relevant variables, namely the crop production at the end of the agricultural season; ($iii$) evaluation of cropping pattern decisions in terms of farmers' payoff at the end of the agricultural season as simulated by the integrated CHNS model. In this step, different levels of risk aversion can be simulated to explore the sensitivity

of the overall assessment with respect to farmers' behavioral attributes.

The first step of the framework (upper block in Figure 2) starts with the post-processing of the hindcast data (see section 3.1) in order to remove the modeling biases generally affecting the simulation of coupled ocean-atmospheric models, such as high rainfall frequency and low rainfall intensity (Ines and Hansen, 2006). The bias-corrected dataset is further downscaled using a stochastic weather generator in order to resolve the spatial and temporal scales mismatch between hindcast data and model

inputs. In particular, the generator allows performing not only the spatial downscaling but also the temporal disaggregation to obtain forecast of daily precipitation and temperature from the forecast products, in case they have a monthly time resolution. The comparison of the post-processed rainfall and temperature forecast products with the on-site historical observations provide a first estimate of the forecast quality.

The post-processed hindcast dataset is then fed into the process-based component of our integrated CHNS model (middle

block in Figure 2). This includes a spatially distributed process-based representation of the Muzza irrigation district (see section 3.2), which extends the assessment of the forecast quality by looking at the difference between forecasted crops' yield and the one simulated using observed time series of precipitation and temperature, assuming the expected crop yield represent the main determinant of farmers' cropping pattern decisions.





The human component of the CHNS (bottom block in Figure 2) is finally introduced in the form of an agent-based decision model (see section 3.3), which allows simulating farmers' cropping pattern decisions driven by different forecast information. This decision model allows coupling the simulation of the process-based model and the prediction of crops' profitability with the selection by each farmer-agent of the best cropping pattern as the one characterized by the highest profitability. The agent-based model allows testing different behavioral criteria, capturing alternative levels of farmers' risk aversion (or degree of trust) with respect to the forecast uncertainty. In particular, we consider a spectrum of behaviors ranging from a fully optimistic farmer, who makes decision on the basis of the best possible situation, to an extremely pessimistic farmer, who, instead, looks at the worst case performance. Then, given the selected cropping pattern by each farmer-agent, the model is simulated using the observed values of precipitation and temperature to obtain the production and the associated profit at the end of the agricultural season. The estimated agents' profit is compared with the one obtained under the hypothesis of perfect foresight, which represents the ideal upper-bound of the system performance. The operational value of WCSs is finally estimated as the percentage of agents making optimal decisions using the forecast products, which represents the opportunity cost of using WCSs with respect to having a perfect foresight. The results are then validated against the profit obtained by the agents when informed with simple empirical forecasts.

Details about each step of the proposed framework, corresponding to a different block in Figure 2, are reported in the next sections.

### 3.1 Post-processing of forecast products

The first step of the proposed procedure (upper box in Figure 2) aims at post-processing the forecast products. Depending on the characteristic of the forecast, we perform bias correction by means of the change factor approach (Crochemore et al., 2016) or the quantile mapping technique (Déqué, 2007).

The quantile-based mapping technique is a statistical downscaling method, which builds the transfer function by mapping the cumulated density function (CDF) of climate model outputs onto the site based observation. The calibrated transfer function is used later on to derive corrected estimates from new incoming outputs by resolving the mismatch between the observed site measurements and the simulated climate outputs. The quantile-based mapping is applied to forecast products providing daily trajectories of precipitation and temperature, which allow a proper estimation of the corresponding CDFs. This step becomes questionable in case of monthly hindcast due to the limited dimension of the dataset. In this case, we apply the change factor approach, in which a multiplicative factor is used to scale the value of precipitation, while an additive factor to adjust the temperature for each month.

Despite the systematic bias in the hindcast dataset can be partially solved by using bias correction, the difficulty in dealing with the uncertainty of ensemble forecasts remains a challenge. Previous studies (e.g., Tippett et al., 2007) have suggested the probabilistic use of long-range weather forecast by deriving the statistical signatures from ensemble forecasts, such as the mean or the anomaly values. This statistic is then compared with the climatology to indicate whether the incoming year is expected to be normal, wet, or dry. As a consequence, the information on the intra-annual variability of the climate,which is critical for crops' growth and agricultural management, is not preserved. Rather, in this work the multi-ensemble data is





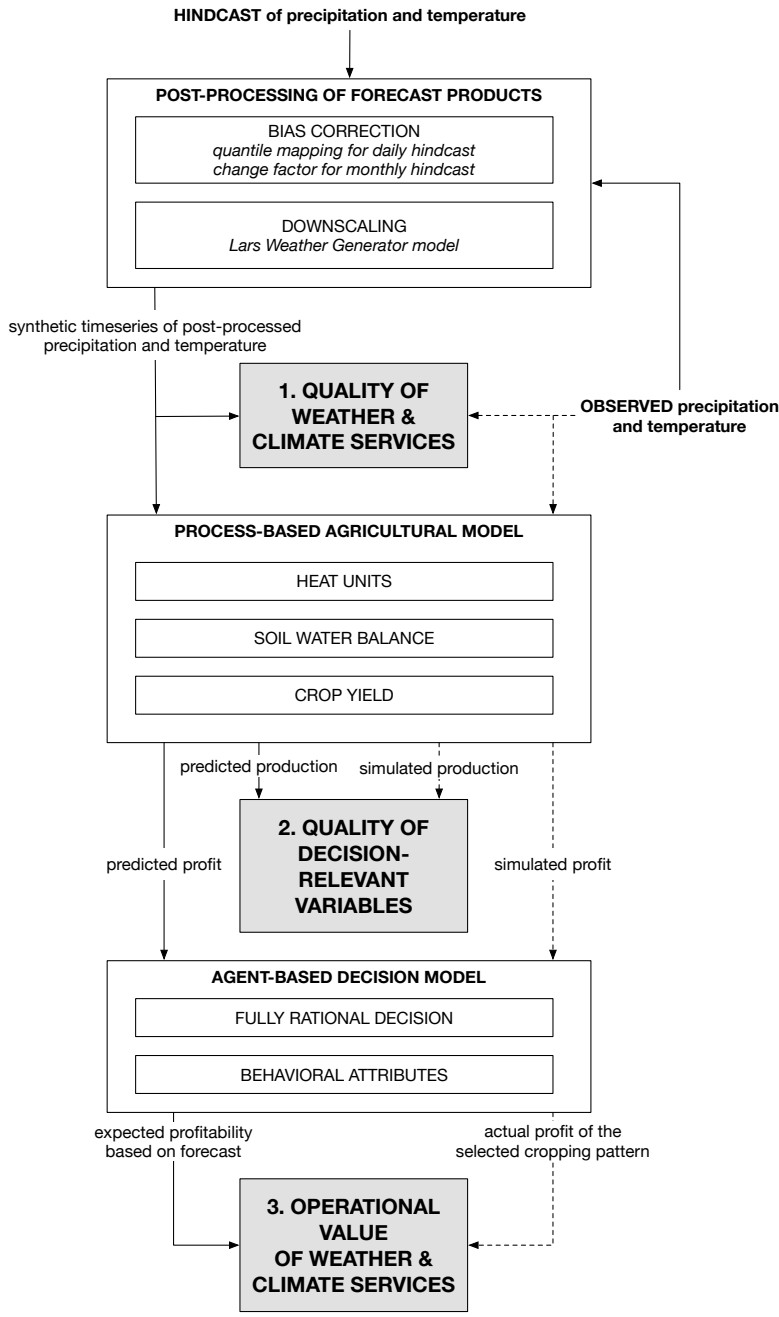

**Figure 2.** Detailed workflow of the proposed three-stage assessment framework of WCSs (solid line) against observed conditions (dashed line).





assimilated into a stochastic weather generator, whose parameters are calibrated from observations and then perturbed based on the forecast conditions. This allows us to generate synthetic time series of precipitation and temperature maintaining the information estimated by the ensemble forecast. In addition, the stochastic weather generator can also disaggregate monthly

forecast into daily values, which are needed to run the process-based model in the next step, without losing the generality of the statistical behavior of the variables. The LARS-WG model (Semenov and Barrow, 1997) is selected for this task as it has been reported to outperform many other weather generators (Hashmi et al., 2011).

The perturbation factors of the mean daily rainfall intensity ($F_i^{P,pert}$) and of the mean temperature ($F_i^{T,pert}$), along with the monthly average number of wet (dry) days ($F_i^{wet,pert}, F_i^{dry,pert}$), are the key parameters of the weather generator. These are

determined according to eq. (1), where $m$ is the number of days in the $i$-th month and $\mathbf{1}(\cdot)$ is the binary operator that returns 1 if the daily precipitation intensity $P_{i,j}$ is larger than 1 mm (wet), and 0 otherwise (dry) (Ceballos et al., 2004). In particular, eqs. (1a)-(1b) represent the expected change of rainfall frequency with respect to the average historical observations ($P^h$) measured by the local stations located in the considered study area, while eq. (1c) specifies the change of rainfall intensity conditioned on the rainy days. For the precipitation, the computed perturbation factor is used to scale up (down) the original parameter

values. The change of monthly mean temperature is formulated in eq. (1d) as an additive term. The ensemble hindcast are aggregated in order to derive a unique factor for each month. The perturbation parameters are then used to generate synthetic, daily time-series for one year according to the considered forecast information.

$$F_i^{\text{wet,pert}} = \frac{\mathrm{E}[\sum_{j=1}^m \mathbf{1}(P_{i,j} \geq 1\mathrm{mm})]}{\mathrm{E}[\sum_{j=1}^m \mathbf{1}(P_{i,j}^h \geq 1\mathrm{mm})]} \tag{1a}$$

$$F_i^{\text{dry,pert}} = \frac{\mathrm{E}[\sum_{j=1}^m \mathbf{1}(P_{i,j} < 1\mathrm{mm})]}{\mathrm{E}[\sum_{j=1}^m \mathbf{1}(P_{i,j}^h < 1\mathrm{mm})]} \tag{1b}$$

$$F_i^{P,\text{pert}} = \frac{\mathrm{E}[P_i|\text{wet}]}{\mathrm{E}[P_i^h|\text{wet}]} \tag{1c}$$

$$F_i^{T,\text{pert}} = \mathrm{E}[T_i] - \mathrm{E}[T_i^h] \tag{1d}$$

## 25   3.2   The process-based agricultural model

The second step of our procedure (middle block in Figure 2) aims at estimating the expected crops' yield at the end of the agricultural season, which is assumed a decision relevant information for the considered farmer-agents. For this purpose, we rely on a spatially distributed process-based model of the Muzza irrigation district (Giuliani et al., 2016), which is composed by three interlaced modules: $i$) a distributed-parameter water balance module that simulate water resources, conveyance, distribution, and soil-crop water balance (Facchi et al., 2004; Gandolfi et al., 2006); $ii$) a heat units module that simulates the




sequence of growth stages as a function of the temperature (Neitsch et al., 2011); $iii$) a crop growth module that estimates the optimal and actual yields, accounting for the effects of water stresses due to the insufficient water supply that may have occurred during the agricultural season (Steduto et al., 2009).

The water balance module partitions the irrigation district with a regular mesh of cells with a side length of 250 m to represent the space variability of crops, soil types, meteorological inputs, and irrigation distribution. Each individual cell identifies a soil volume which extends from the soil surface to the lower limit of the root zone. This soil volume is subdivided into two layers, modeled as two non-linear reservoirs in cascade: the upper one (evaporative layer) represents the upper 15 cm of the soil; the bottom one (transpirative layer) represents the root zone and has a time-varying depth. The water percolating out of the bottom

layer constitutes the recharge to the groundwater system.

The heat units module defines the relationships between the temperature and some variables and parameters related to the crop growth stage (e.g., root length, basal coefficient, leaf area index), which also influences the water balance module. According to the heat units theory (Neitsch et al., 2011), crop growth stage at time $t$ in the $i$-cell is defined as a function of the cumulated heat units ($HU_t^{(i)}$). A range is defined for each crop: the minimum is the base temperature $T_b$, which defines the

day of sowing (i.e., when $HU_t^{(i)} > T_b$) and the maximum is the cut-off temperature over which the heat units are no longer cumulated.

Finally, the crop growth module first estimates the maximum yield achievable in optimal conditions and, then, reduces it to take into account the stresses due to insufficient water supply from rainfall and irrigation happened during the agricultural season. The yield response to water stresses is estimated according to the empirical function proposed in the AquaCrop model

(Steduto et al., 2009; Raes et al., 2009) and based on the approach proposed by FAO (Doorenbos et al., 1979):

$$1 - \frac{Y_{real}^{(i)}}{Y_{opt}^{(i)}} = k_y \left( 1 - \frac{Tr_{real,tot}^{(i)}}{Tr0_{tot}^{(i)}} \right) \tag{2}$$

where $Y_{real}^{(i)}$ and $Y_{opt}^{(i)}$ are the actual and optimal yield in the $i$-th cell, $Tr_{real,tot}^{(i)}$ and $Tr0_{tot}^{(i)}$ the actual and optimal transpiration in the $i$-th cell during the whole growth period, and $k_y$ is a crop-specific coefficient relating yield decline and water stress.

## 3.3  The agent-based decision models

In the last step of our procedure (bottom block in Figure 2), the process-based model described in the previous section is combined with an agent-based model representing the decisions made by the farmers in the 39 irrigation units of the Muzza irrigation district. In particular, each irrigation unit is modeled as a single agent and the decision of each agent is limited to a single crop in each agricultural season. The possible crop choices include tomato, corn, soybean and rice, which represent the most common crops in the considered study area. The crop growing period slightly varies from one crop to the other, with maize being the crop with longest growing period (see Figure 3). Note that the modeled agents do not represent individual farmers in the system, but rather a group of farmers located in one of the 39 irrigation units. This hypothesis is tantamount





to describe the median behavior of the ensemble of farmers aggregated at the irrigation unit level under the assumption of rational behaviors, and provides a simple and effective way to capture the inter-annual dynamics of land use at the district scale (Giuliani et al., 2016).

The agent's decision problem is hence formalized as follows:

$$\gamma_k^* = \arg\max_{\gamma_k} \Psi_{\hat\varepsilon} \left[ \mathcal{P}\left( Y_{real}(\gamma^k), p(\gamma^k), c(\gamma^k), \sigma(A_k) \right) \right] \qquad k = 1, \ldots, 39 \tag{3}$$

where $\mathcal{P}(\cdot)$ is the net profit obtained at the end of the agricultural season from the yield $Y_{real}(\gamma^k)$ of crop $\gamma^k$ (estimated from eq. 2), $p(\gamma^k)$ and $c(\gamma^k)$ are the corresponding price and cost, respectively, and $\sigma(A_k)$ the subsides for the $k$-th agent (with $k = 1, \ldots, N$). The subsides, which depend on the cultivated area $A_k$ and not on the selected type of crop (Gandolfi et al., 2014), derive from the EU's Common Agricultural Policy (CAP), which complements a system of direct payments to farmers with measures to help rural areas in facing a wide range of economic, environmental, and social challenges (Britz et al., 2003).

In Problem (3), the optimal cropping pattern decision $\gamma_k^*$ is conditioned on the forecast information $\hat\varepsilon$, with the statistic $\Psi$ filtering the uncertainty in the forecast products and capturing the personal risk aversion of each farmer-agent (Giuliani and Castelletti, 2016). In fact, depending on its personal behavioral attitude and on its level of trust in the forecast products, an agent can use the forecast information in different ways, particularly when this is provided in the form of prediction ensembles. In this work, we explore three different levels of farmers' risk aversion creating a spectrum of behavioral attitudes, namely risk averse, risk neutral, or risk prone (e.g., Rogers, 1975; Mosley and Verschoor, 2005; Koundouri et al., 2006; Djanibekov and Villamor, 2017). A risk averse, pessimistic behavior (or a low level of trust in the forecast products) implies that agents decide on the basis of the worst-case realization, which means they will select the cropping patterns able to ensure the highest profit in the most adverse conditions. Yet, these decisions may result to be overly conservative if the actual realization is different from the worst possible one. Conversely, a risk prone, optimistic behavior produces decisions that rely on the best possible situation. This choice, however, increases the risk of cultivating crops that are highly productive under favorable weather conditions, but might be also highly vulnerable under more adverse seasons. Finally, risk neutral agents with a sufficient level of trust in the forecasts' products ground their decisions on the expected profitability of the crops using the probability of realizations derived from the forecast information.

These alternative behaviors are formalized by means of the following three statistics $\Psi$ which are used in eq. (3) to filter the uncertainty in the forecast products:

- Risk averse behaviors are modeled using the *minimax regret* metric (Savage, 1951), where decisions are based on the regret, defined as the difference between the performance resulting from the best alternative given that the predicted $\hat\varepsilon_j$ is the true realization of precipitation and temperature, and the performance of a given cropping pattern $\gamma$ under the same weather conditions $\hat\varepsilon_j$, i.e.

$$r(\gamma, \hat\varepsilon_j) = \max_{\gamma} \left( \mathcal{P}(\gamma, \hat\varepsilon_j) \right) - \mathcal{P}(\gamma, \hat\varepsilon_j) \tag{4}$$





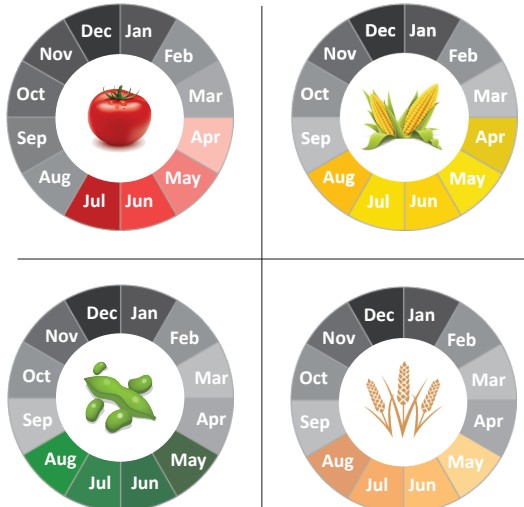

**Figure 3.** Growing period of the four considered crops.

Then, this metric selects the best cropping pattern $\gamma^*$ adopting a pessimistic approach, namely by minimizing the maximum regret across all the members of the forecast ensemble $\hat{\varepsilon} \in \Xi$, i.e.

$$\gamma^* = \arg\min_{\gamma} \left( \max_{\hat{\varepsilon} \in \Xi} r(\gamma, \hat{\varepsilon}) \right) \tag{5}$$

20    – Risk neutral behaviors are modeled using the *principle of insufficient reason* (Laplace, 1951), where decisions are made by assigning equal probability to each forecast ensemble member. Then, the best cropping pattern $\gamma^*$ is selected as the one associated to the maximum expected performance, i.e.

$$\gamma^* = \arg\max_{\gamma} \left( \frac{1}{n} \sum_{j=1}^{n} \mathcal{P}_j(\gamma, \hat{\varepsilon}_j) \right) \tag{6}$$

where $n$ is the number of members in the ensemble.

25    – Risk prone behaviors are modeled using the *maximax* metric (French, 1988), where decisions are made by looking at the best possible performance of each decision and selecting the cropping pattern $\gamma^*$ such that

$$\gamma^* = \arg\max_{\gamma} \left( \max_{\hat{\varepsilon} \in \Xi} \mathcal{P}(\gamma, \hat{\varepsilon}) \right) \tag{7}$$

This metric is generally associated with an optimistic point of view as it assumes that the best state of the world will realize.





| Institutes | Model Name | Spatial Resolution (°) | Temporal Resolution | Lead-time | Reference |
|---|---|---|---|---|---|
| ECMWF | IFS/HOPE | 2.5×2.5 | daily | 7-month | Facchi et al. (2004) |
| ECMWF | ECHAM5/MPIOM | 2.5×2.5 | daily | 7-month | Roeckner et al. (2003) |
| ECMWF | HadGEM2-AO | 2.5×2.5 | daily | 7-month | Martin et al. (2011) |
| ECMWF | IFS/HOPE | 2.5×2.5 | daily | 14-month | Facchi et al. (2004) |
| ECMWF | ECHAM5/MPIOM | 2.5×2.5 | daily | 14-month | Roeckner et al. (2003) |
| ECMWF | HadGEM2-AO | 2.5×2.5 | daily | 14-month | Martin et al. (2011) |
| ECMWF | IFS/HOPE | 2.5×2.5 | monthly | decadal | Facchi et al. (2004) |
| ECMWF | ECHAM5/MPIOM | 2.5×2.5 | monthly | decadal | Roeckner et al. (2003) |
| ECMWF | HadGEM2-AO | 2.5×2.5 | monthly | decadal | Martin et al. (2011) |
| ECMWF | DePreSys | 2.5×2.5 | monthly | decadal | Liu et al. (2012) |
| NCEP | CFS v2 | 0.9375×0.9375 (i.e., T126 Gaussian) | 6-hourly | 9-month | Kim et al. (2012) |
| CCCma | CanSIPS CamCM3 | 2.5×2.5 | monthly | 12-month | Kharin et al. (2009) |
| CCCma | CanSIPS CamCM4 | 2.5×2.5 | monthly | 12-month | Kharin et al. (2009) |
| Empirical | EmpPast | - | - | - | - |
| Empirical | Emp2Ave | - | - | - | - |
| Empirical | EmpClima | - | - | - | - |

**Table 1.** Summary of the WCSs used in this work. The entries from 1 to 10 are all obtained from ECMWF ENSEMBLE project with different lead-time.

## 4 Experiment settings

Hindcast of precipitation and surface temperature data are collected from the ECMWF Stream 2 project, NCEP, and Canadian Centre for CCCma, respectively. Table 1 reports some general information about the considered forecast products.

The ECMWF hindcast consists of a comprehensive set of seasonal, annual, and decadal products. The Climate Forecast System version 2 (CFS v2) from NCEP is similar to ECMWF products, generated using fully coupled models representing the interactions between the Earth's atmosphere, oceans, land and sea-ice (Saha et al., 2014). The Canadian Seasonal to Inter-annual Prediction System (CanSIPS) is a long-range multi-model prediction system whose objective is to forecast the evolution of global climate conditions (Merryfield et al., 2011). There are two versions of coupled climate models inside the CanSIPS system, namely the CamCM3 model (Arora et al., 2011) and CamCM4 model (Scinocca et al., 2008). To tackle with the impact of uncertainties in the initial conditions, most models run a number of simulations with slightly different atmospheric and oceanic initial states to generate ensemble outputs.

In addition to the institutional forecast products, we also include in the analysis three simple empirical models representing farmers' prior knowledge based on past observation. Specifically, EmpPast refers to the empirical forecast obtained by duplicating the past year's observations. The Emp2Ave stands for the simple forecast averaging the past two years' observations





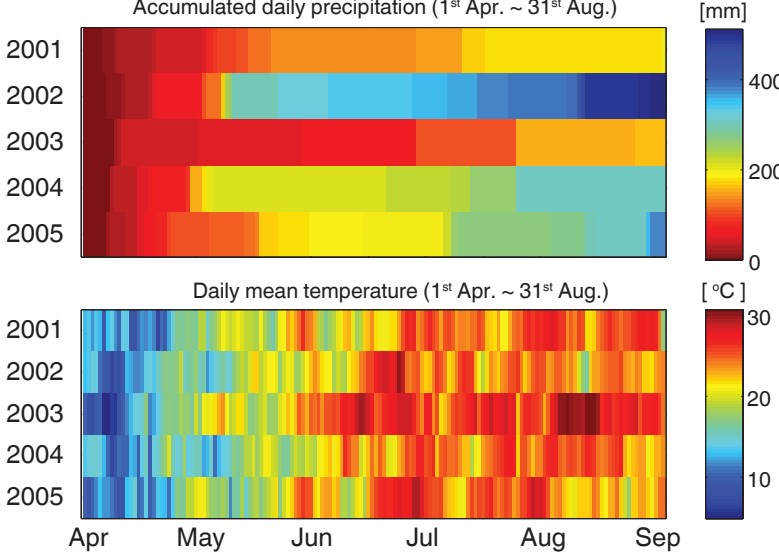

**Figure 4.** Observed precipitation and temperature from 2001 to 2005. The time-series starts from $1^{st}$ April till the end of August covering only the crop growing period.

which is analogous to the climatology forecast with a 2-year memory basis as reflective of farmers' best possible capacity. Lastly, the EmpClima is simply the climatology forecast over past observations.

A total 13 years' observations (1993-2005) are available, with the last five years (i.e., 2001-2005) used for running the model simulations and estimate the operational value of the considered forecast products. This horizon was selected to include a fairly balanced number of normal, wet, and dry agricultural seasons with variable temperature patterns, as shown in Figure 4. For each simulation, all observations available at the beginning of the year are used for bias correction and downscaling. For example, if the simulation starts in 2003, then the control dataset from 1993 to 2002 is used to calibrate the transfer function and the parameters in the LARS-WG, which will be used to bias correct the hindcast data in 2003 and generate the time-series of precipitation and temperature required by the model. As an illustrative example, Figure 5 shows the performance of bias-correction for IFS/HOPE model from ECMWF annual product in 2002. The hindcast data (top panels) clearly overestimate the rainfall frequency and underestimate the rainfall intensity, particularly during the summer. The bias correction successfully solves these issues and the post-processed values match the observations (bottom panels).

## 5 Numerical results

The first step of our framework (Figure 2) aims at evaluating the forecast quality in terms of the difference between the post-processed forecast variables and the observed ones. Figure 6 shows the post-processed forecast of rainfall against the observed





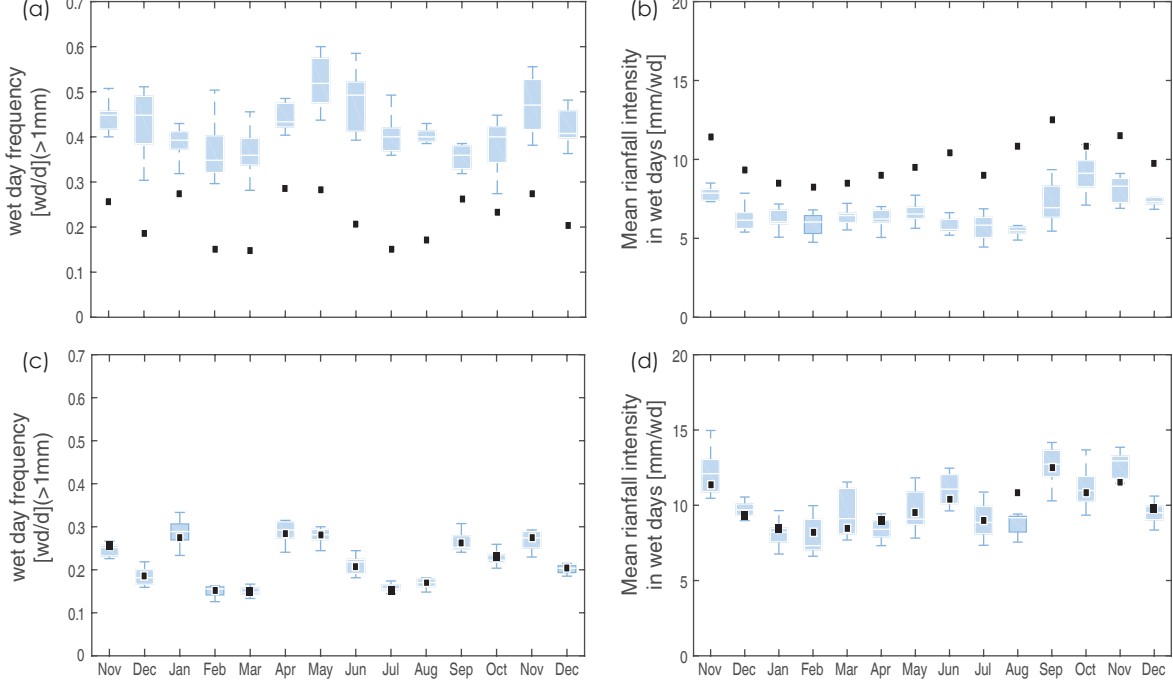

**Figure 5.** Comparison between the mean rainfall intensity and wet day frequency for IFS/HOPE model from ECMWF annual product in 2002 before (panels a-b) and after (panels c-d) the bias-correction

one during the crop growing season across different forecast products. The empirical memory based forecast (blue bars in Figure 6) assumes the weather in the incoming year is similar with previous conditions. This mechanism leads to significant forecast errors, such as 2002, which was predicted as a normal year but it was wet, or 2003, which was predicted as wet but

5   was extremely dry. The climatology-based forecast assumes the realization of average conditions determined from historical observations. This strategy works in normal years, such as 2001 or 2004, while extreme weather conditions tend to be filtered out during the averaging and year-to-year variations are less significant. For the institutional forecast products, CFS product seems to work well in normal years while being less accurate in wet and dry years. In particular it is not able to capture the variation from high to low rainfall in 2002-2003. Similar results can be observed for Canada CanSIps products, with CamCM4

10   generally underestimating more the rainfall compared with the CamCM3. Estimating the total rainfall for wet/dry years is challenging also for ECMWF products, which involve multiple forecast systems at various lead time. Nevertheless, there are some exceptions, such as IFS/HOPE model from annual forecast products (red-edge marked bars), which seems to be able to predict quite well the variability from 2002 to 2004. Similarly to the results in Figure 6, the comparison of the forecast quality evaluated with respect to the daily mean temperature reported in Figure 7 shows similar patterns among all products, with





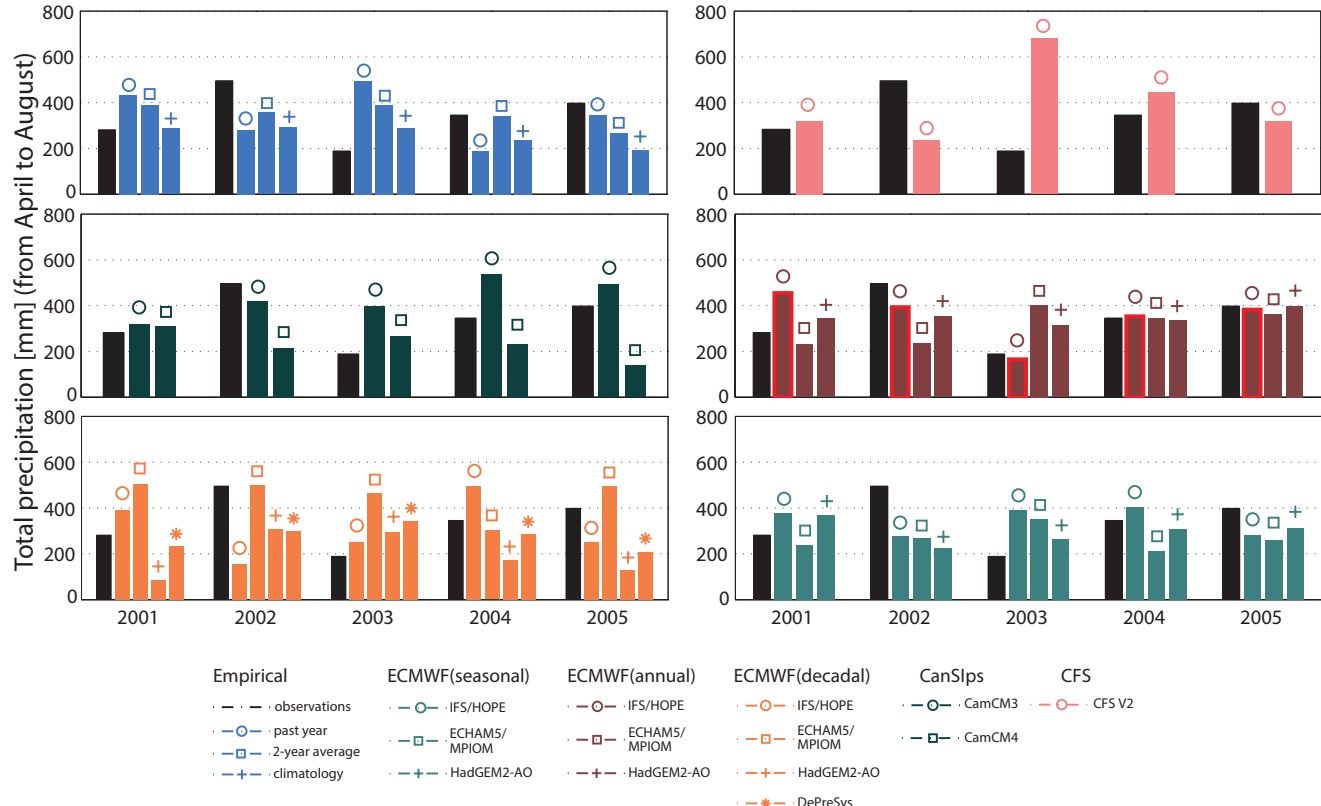

**Figure 6.** Comparison of predicted and observed total rainfall during the crop growing season across different forecast products. The color is associated with the family of forecast products and the markers on the top of the bars represent different models within a single family.

most of the forecast values that are close to the observed one except for 2003, which was an extremely hot summer due to the European heat wave (García-Herrera et al., 2010).

The second step of our framework (Figure 2) moves the focus from climatic variables to decision-relevant variables obtained via simulation of the process-based agricultural model. In our problem, this implies looking at the accuracy of estimating the

5    crop yield given the post-processed forecast information. The agricultural model described in section 3.2 provides a mean to transform the climatic variables of interest into the crop yield, which is considered one of the most relevant variables involved in farmers' decision-making process. The comparison between the predicted and the observed (i.e., simulated using the observed weather data for removing possible model biases and focus only on the forecast errors) crop productions is reported in Figure 8. In general, the fluctuations of the production follow the fluctuations of climate variables, especially

10   the precipitation, with the highest productions in 2002 and the lowest in 2003. For most institutional forecast products, the







**Figure 7.** Comparison of predicted and observed daily mean temperature during the crop growing season across different forecast products. The color is associated with the family of forecast products and the markers on the top of the bars represent different models within a single family.

predicted crop productivity in wet/dry years is significantly different from the ones obtained with the observed climate. In many cases, several products tend to overestimate crop yield in dry years and to underestimate it in favorable wet years. One exception is again represented by the IFS/HOPE model, which is able to provide quite accurate forecast of crop yield. These results suggest that, as expected, forecasting crop yields is a more complex task than forecasting precipitation and temperature. 5 This is further confirmed by the poor performance attained by the empirical products when their forecast quality is evaluated in terms of crop yield. Especially for water demanding profitable crop like tomato, erroneously forecasting a wet year causes an over-optimistic expectation, which significantly differ from the actual outcome. Similarly, some products (e.g., decadal forecast from ECMWF ECHAM5/MPIOM models) may forecast a wet year which instead results to be dry, such as 2005, and



produce a large overestimation of crops' productivity. Finally, these results also show the emergence of some differences in the accuracy of precipitation and temperature forecasts with respect to the corresponding prediction of crop yield. A clear example is 2001, for which CFS V2 exhibits a significantly higher accuracy in predicting the precipitation than IFS/HOPE model with ECMWF annual product (see Figure 6). Yet, this superiority does not imply a better forecast of the crops' production and both

the products indeed have similar levels of accuracy across all the four simulated crops (see Figure 8).

Looking at the accuracy of the predicted precipitation and temperature as well as the predicted crop yields provides a measure of the forecast quality without exploring the potential benefit of using WCSs to inform the farmers' decisions. The quantification of the operational value of WCSs is performed in the third step of our framework (Figure 2), where we use our agent-based model to simulate farmers' decision-making process and estimate the profit obtained from the cultivation

of the selected crops. This is contrasted with the profit obtained under the assumption of perfect foresight to estimate the opportunity cost of using WCSs. It is worth pointing out that, although perfect forecast accuracy can be hardly achieved, farmers' decisions under forecast information may coincide with that selected with the perfect foresight. Figure 9 illustrates the relationship between the performance of agents decisions (x-axis), measured in terms of fraction of farmers making optimal decisions (i.e., selecting the same cropping pattern as in the perfect foresight case) and attaining an opportunity cost equal

zero, and the associated forecast quality (y-axis), evaluated in terms of mean absolute error (MAE) of the selected crops. The scatterplot is divided into four zones, where the bottom right corner indicates that a good prediction skill leads to better decision outcomes, while the upper left corner corresponds to the situation where forecast errors induce a large opportunity cost. Both the empirical forecast and the institutional forecast products are spread along the y-axis, confirming the variability of forecast quality in predicting crops' productivity. Numerical results show that most of the points characterized by a good

forecast quality, defined as MAE below 1000 kg/ha, corresponds to institutional products. These high quality forecast products provide valuable information to support agents decisions, as demonstrated by the fact that all the points below the 1000 kg/ha line successfully inform a large fraction of agents (i.e., 90%-100%), who are able to make optimal decisions. However, many empirical products are also able to achieve zero opportunity cost, even though their forecast quality is generally worse than the one of institutional forecasts. This can be explained by considering that agents are deciding looking at the ranking of

crops' profitability rather than on their absolute expected profitability. As a consequence, an overall under/overestimation of the profitability of all the crops (e.g., the profit of each crop is predicted to be 10% lower than reality) results in a poor forecast quality but, at the same time, this forecast error does not generate a rank reversal and the agents select the optimal cropping pattern anyway.

## 6   Impacts of farmers' behavioral attitudes

The results presented in the previous section are obtained assuming risk-neutral agents, where the most profitable cropping pattern is selected by the modeled agents on the basis of the crops' profitability predicted by the agricultural model when simulated under a single synthetic timeseries of post-processed precipitation and temperature (see Figure 2). Yet, in a more realistic setting, farmers are exposed to uncertain forecasts and, moreover, their behavioral factors may influence the use of

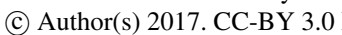





**Figure 8.** Simulated crop productions from 2001 to 2005. The arrangement of simulated crop types are tomato (upper left), maize (upper right), soybean (lower left) and rice (lower right).





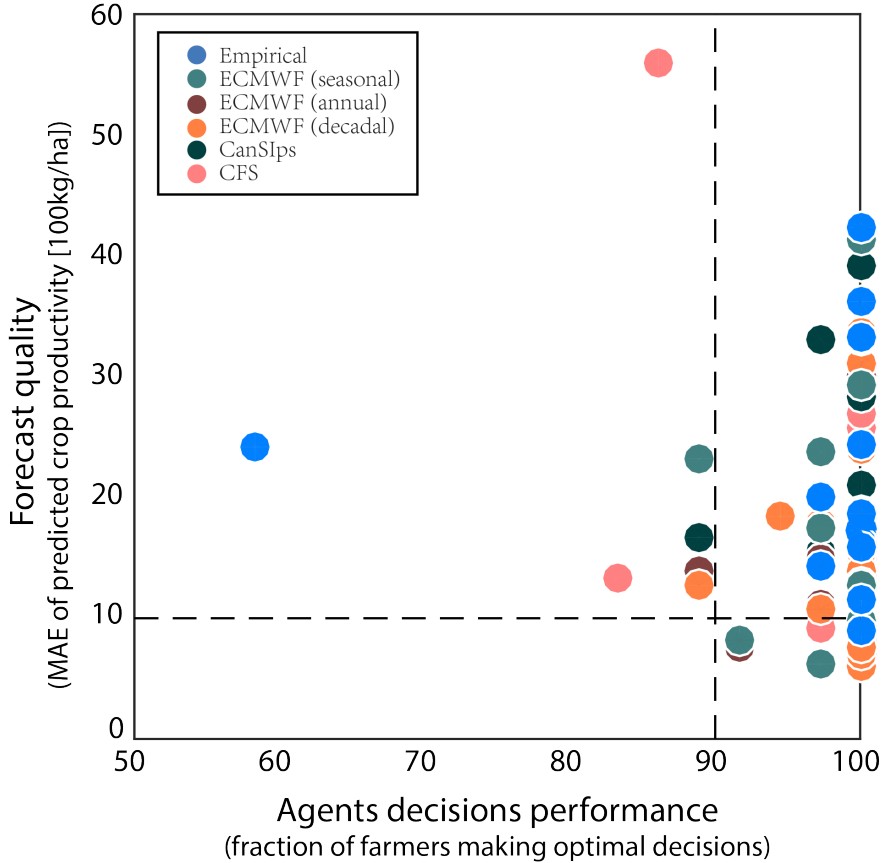

**Figure 9.** Scatterplot of forecast quality of predicted crop productivity (y-axis) and farmers' crop decisions performance (x-axis) under different forecast products.

WCSs. In this section, we explore how different levels of risk aversion impact on the agents' decision and, consequently, on the estimated operational value of WCSs. In particular, we focus on the ECMWF annual product, which attained both the high forecast quality and high operational value, and we generate 100 synthetic time series of precipitation and temperature by means of the weather generator. These simulations yield 100 uncertain values of crops' profitability at the end of the agricultural

5   season. This uncertainty is then filtered by agents through a proper statistic capturing their personal risk aversion, including risk neutral, risk prone, and risk averse behaviors (see section 3.3).

   The results obtained adopting these different levels of risk aversion are reported in Figure 10, where the left y-axis shows the distributions of the forecast quality for the three considered models, while the right y-axis shows the fraction of farmer-agents making optimal decisions. The figure shows that, although we are considering a single forecast product, the forecast quality

10   varies according to the model used for producing the forecast, with IFS/HOPE characterized by the lowest MAE, both in terms




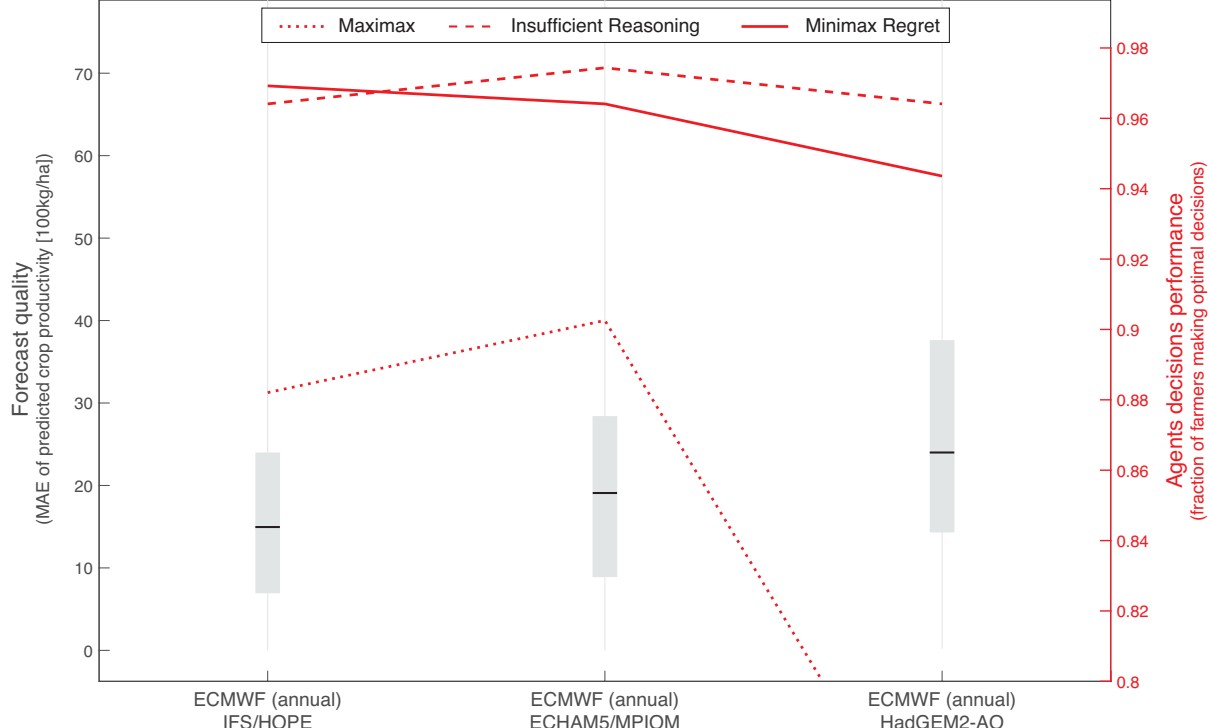

**Figure 10.** Forecast quality uncertainty of predicted crop productivity (left y-axis) and farmers' crop decisions performance for different levels of risk aversion (right y-axis).

of median and variance, and outperforming both ECHAM5/MPIOM and HadGEM2-AO. Interestingly, these differences in terms of forecast quality are not linearly transferred to the performance of agents' decisions. Our results show that the level of agents' risk aversion significantly impact on their use of forecast products. Risk averse behaviors (i.e., agents deciding on the basis of minimax regret, represented by the solid red line) attain a performance that decreases when moving from high to low

5     quality forecast. However, this does not hold for risk neutral or risk prone behaviors, simulated as agents deciding according to the principle of insufficient reason (red dashed line) and the maximax metric (red dotted line), respectively. In both cases, the highest fraction of agents making optimal decisions is obtained by using the ECHAM5/MPIOM forecast despite this product has a lower quality than IFS/HOPE.

Finally, it is worth noting that the criterion associated to the largest fraction of agents making optimal decisions, which might

10    be considered as the "best" way for taking advantage of WCSs, varies across the models. The minimax regret is the best when applied to IFS/HOPE forecast, while the principle of insufficient reason is superior when used for ECHAM5/MPIOM and HadGEM2-AO products. A mis-definition of the stakeholders' perception of WCSs, here explored in terms of risk aversion, may hence represent a strong bias in the analysis of WCSs operational value. For example, the opportunity cost of using ECHAM5/MPIOM simulated assuming the principle of insufficient reason is equal to 3%, meaning that one agent over the





39 considered in our model is selecting a sub-optimal cropping pattern. The opportunity cost for the same product simulated assuming the minimax regret is instead equal to 4%, meaning that two agents over 39 select sub-optimal cropping patterns.
Finally, the simulation of risk prone agents adopting the maximax criterion produces an opportunity cost of 10%, meaning that four agents select sub-optimal cropping patterns. These results provide strong evidence about the importance of considering personal, behavioral attributes to produce a proper assessment of WCSs operational value.

## 7   Conclusions

In this work, we propose a novel framework for assessing the operational value of several Weather and Climate Services. This
approach, which relies on an integrated model of a Coupled Human-Natural System, is applied in the Muzza irrigation district (Italy), a complex agricultural system where farmer-agents select the crops to cultivate by maximizing the expected net profit at the end of the agricultural season. Our framework allows quantifying the quality of the considered forecast products both in terms of climatic and decision-relevant variables as well as estimating the associated payoff for the farmers, also exploring the impacts of behavioral attributes on the uptake and use of WCSs.
Our study shows that, at present, the accuracy of most state-of-the-art weather forecast products is still limited, especially in the prediction of precipitation with a lead-time of 7 months or longer. The ECMWF annual forecasts simulated by the IFS/HOPE model displayed the maximum forecast skill among the considered products and they were able to also predict some extreme events, including the intense drought of 2003. The predictions of crop yield obtained via simulations of process-based models using the predicted values of precipitation and temperature as climate forcing show similar performance in terms
of forecast quality.

Numerical results on the use of these forecast to inform agents' decisions show that the accuracy of estimating crop yield and the probability of making optimal decisions are not necessarily linearly correlated. The assessment of the operational value of WCSs should therefore include a decision model reproducing the actual users' adoption of forecast products within their decision making process. Some institutional forecast (e.g., ECMWF products) attain both high forecast quality and high agents
decisions performance. However, our results also show that in many cases the agents decisions are still optimal even though informed by products with low forecast quality (e.g., CFS products). From the farmers point of view, the operational values of ECMWF and CFS products are therefore equivalent despite ECMWF would largely outperform CFS in terms of forecast quality. Finally, we provide numerical evidence about the impacts of different farmers' behavioral attributes (i.e., levels of risk aversion) on the quantification of WCSs operational value. The exploration of this behavioral uncertainty further amplifies the key role of the decision model in the assessment procedure. Our results show that the opportunity cost of the same forecast product increases from 3% to 10% while moving from risk neutral to risk prone decisions, potentially producing rank reversals
in the quantification of the WCSs operational value.

Future research efforts will focus on better understanding how to better tailor WCSs information to support stakeholders' decisions (van den Hurk et al., 2016). In addition, we will analyze the diffusion dynamics of the use of these forecast products




after the adoption by a small group of innovative individuals, which will then influence the behaviors of other individuals (Young, 2011).

*Acknowledgements.* The work has been partially funded by the IMPREX project funded by the European Commission under the Horizon 2020 framework programme (grant n. 641811).





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
