# Peer review of "A coupled human-natural system to assess the operational value of weather and climate services for agriculture"

_Hydrology and Earth System Sciences, 2017_

## Referee Comment (RC1) · Anonymous Referee #1 · 15 Jun 2017

**GENERAL COMMENTS**

The subject of the paper "*A coupled human-natural system to assess the operational value of weather and climate services for irrigated agriculture*" is of direct interest to the Journal of Hydrology and Earth System Sciences. Authors introduce and apply a framework in the context of measuring the operational value of weather and climate services (WCs). The validation of the usefulness of the WCs to the final users is a much needed step towards the realization of these services.

Regarding the different aspects of the HESS journal:

| | | |
|---|---|---|
| 1 | Does the paper address relevant scientific questions within the scope of HESS? | YES |
| 2 | Does the paper present novel concepts, ideas, tools, or data? | YES |
| 3 | Are substantial conclusions reached? | YES |
| 4 | Are the scientific methods and assumptions valid and clearly outlined? | YES (see technical comments) |
| 5 | Are the results sufficient to support the interpretations and conclusions? | YES |
| 6 | Is the description of experiments and calculations sufficiently complete and precise to allow their reproduction by fellow scientists (traceability of results)? | YES (I also encourage the authors to share the code/data) |
| 7 | Do the authors give proper credit to related work and clearly indicate their own new/original contribution? | YES |
| 8 | Does the title clearly reflect the contents of the paper? | YES |
| 9 | Does the abstract provide a concise and complete summary? | YES |
| 10 | Is the overall presentation well structured and clear? | YES |
| 11 | Is the language fluent and precise? | YES |
| 12 | Are mathematical formulae, symbols, abbreviations, and units correctly defined and used? | YES |
| 13 | Should any parts of the paper (text, formulae, figures, tables) be clarified, reduced, combined, or eliminated? | YES (see technical comments) |
| 14 | Are the number and quality of references appropriate? | YES |
| 15 | Is the amount and quality of supplementary material appropriate? | # |

**SPECIFIC COMMENTS**

1. One of my concerns is the limited duration of the analysis period (2001-2005). Why authors didn't extend the analysis beyond 2005. Is it due to the limited data availability? If yes, it would be also interesting to see similar results for a longer time period even for less forecast products.

2. Assuming that instead a single forecast product, a large ensemble developed by the combination of several products could outperform the forecast quality or the result to better decisions compared to single products?

3. Another subject that could also be discussed is the limitations and/or assumptions of the study. I think that a limitations section should be added in the paper in order to summarize the main simplifications or assumptions considered in the work. For example the determinant yield factor is the water availability no matter the agricultural treating of the farmers during the cultivation period. Maybe such a section could also include some references to works in which they have been treated in other way.

Considering these and the fact that the scientific significance and quality are excellent, my suggestion to the editors would be to accept after minor revision in the context of my specific and technical comments. I am listing a number of suggestions in the form of technical comments that will improve the presentation of the study.

*TECHNICAL COMMENTS*

P4 – Study site section: since you are dealing with end-user services it would be nice if you include more information, a short description of the users (total number of farmers, average farm extent, etc.).

P4 – L9: Here you mention 40% for maize while in Fig. 1 shows 74%.

P5 – L2: you could also say that climate change has exacerbated the severity of the extreme events (drought/heat wave).

P5 – L5: what about 2001? Judging from Fig. 4 2001 was even drier than 2003 and 2005 (also in Fig. 6 for the April to August precipitation).

P5 – L26: add cross reference for Table 1.

P9 – L5: Why do you set the resolution to 250m? Is this resolution adequate for representing the spatial detail of the crops/properties?

P9 – L25: Does this model take into account the behavioral dependency on the preceding year? Meaning that the farmers' decision is affected for example from a "previous (i-1) dry year" and as a result the potentially optimistic decision of year i would be more pessimistic?

P12 – Table1: the products listed here are single member experiments or there is a number of realizations?

P13 – Figure 4: You could also add total precipitation and average temperature for each year (row) on the right part of the figure (on the left from the legend).

P15 – Figure 6: it would be easier to read if you place the legend of each product on the corresponding sub-plot. Otherwise you could arrange the legend in similar order as the subplots because it is hard to detect. It would be also helpful if you could highlight the dry years.

P16 – Figure 7: The differences are hard to distinguish. You could plot the anomalies instead or adjust the range of the temperature axis (for example from 17 to 23ºC). Again it would be also helpful if you could highlight the dry years.

P18 – Figure 8: You could use a continuous line for the deterministic simulation.

P23 – Line 32: remove the space from "f armers"

---

## Referee Comment (RC2) · Anonymous Referee #2 · 20 Jun 2017

The presented manuscript describes and applies a methodological framework to assess the operational value of weather and climate forecast products on irrigated agriculture. It combines a set of forecast products with an agronomic model that simulates the crop yield based on meteorological inputs and an agent-based model that establishes the optimal cropping pattern depending on the forecasts available and the risk profile of the farmers. The novelty of the paper consists in the joint assessment of the forecast quality and its impact on management decisions and farmers risk profile. The methodology is well described and the structure and organization of the paper is coherent and adequate. The results point at the fact that the forecast quality is not necessary correlated with its impact on management decisions.

The paper fits the scope of the journal and has a clear potential for publication, given the increasing momentum of weather and climate services and how its "real" impact can be measured. I have no major concerns about the manuscript, although some improvements would further increase its quality. Therefore, I would consider it ready for publication after fixing the minor concerns I point at below.

TECHNICAL COMMENTS

1. Page 2, lines 28-31: In my opinion, the first sentence of this paragraph is just a summary of the previous one. I would delete it and reflect in the previous paragraph that an alternative promising metric would be the quality obtained on predicting decision-relevant variables.

2. Page 3, line 23: Although it becomes clear when moving forward that "post-processing" means "downscaling and bias-correction", I would add a remark here just to clarify it.

3. Page 4, line 6: what do you mean when you state "pilot"? I think it is a synonym of "case study", but sometimes the term "pilot" implies you run field experiments to apply the method developed. Please clarify the term.

4. Page 5, lines 33-34: can you provide information to support the assumption of using crop yield as main driver of the cropping pattern decisions? Sometimes other variables like management complexity or profit predictability is more important than crop yield. In my opinion, you should clarify, if it is the case, that you make this assumption in the absence of more detailed information about the farmers' decision-making process.

5. Page 6, line 21: as far as I know, the quantile-based mapping is a bias correction procedure. It is true that it has some downscaling component due to matching CDFs obtained at different spatial scales but, on a broader view, it is considered as a bias correction technique. In fact, you previously named it as a bias correction technique. Please fix this.

6. Page 8, lines 15-16: the way in which the aggregation is performed it is not clear. I assume you aggregate the daily data of the same month, but it may also mean you aggregate the ensemble members. Please clarify it.

If you aggregate the ensemble members to obtain a unique factor, I would rather suggest keeping the factor obtained by each ensemble member and generate synthetic daily time series with all of them. In this way, you will have a better representation of the extremes, which are flattened when taking the average.

7. Page 12, table 1: please include the ensemble members of each WCS used unless all the products provide just one ensemble. In this last case, you should indicate in the text that all of them provide a unique ensemble member.

8. Page 15, figure 6: from my point of view, the understandability of this figure would be increased by including the legend inside each individual plot as well as the name of the WCS product. Otherwise the reader needs to constantly go up and down the figure to find out what each bar refers to.

9. Page 16, figure 7: same comment as for figure 6.

10. Page 18, figure 8: I would include the name of the WCS product in each individual plot. Furthermore, I would also provide the value of an average score for the time series inside each plot (for example the MAE). In this way, the reader has a numerical way to easily compare the accuracy of each WCS product type for each plot.

11. Page 19, lines 1-5: Did you generate 100 time series for each year between 2001 and 2005? Did you choose one year between 2001 and 2005 and then generate 100 series for it? Or did you spare the 100 time series between 2001 and 2005? Please add a clarification about it.

12. Page 20, lines 1-14: In my opinion, the fact that the neutral or optimistic risk profiles did not obtain the best performance for the best forecast deserves more explanation. How can you justify this issue? In the absence of more information, I would doubt about

the suitability of the score used (median and variance of MAE). Maybe the IFS/HOPE product does not predict extremes as ECHAM5/MPIOM does, and due to this reason the latter offers the best performance on both the neutral and the optimistic risk profiles. Please add some explanation or theory about this unexpected finding.

---

## Author Comment (AC2) · 14 Jul 2017

We thank the reviewer for the positive comments and thorough review that will surely help us improving the manuscript. We will take all of them into consideration while revising the paper. Below, our point-to-point response.

Referee comment n. 2

The presented manuscript describes and applies a methodological framework to assess the operational value of weather and climate forecast products on irrigated agriculture. It combines a set of forecast products with an agronomic model that simulates

the crop yield based on meteorological inputs and an agent-based model that establishes the optimal cropping pattern depending on the forecasts available and the risk profile of the farmers. The novelty of the paper consists in the joint assessment of the forecast quality and its impact on management decisions and farmers risk profile. The methodology is well described and the structure and organization of the paper is coherent and adequate. The results point at the fact that the forecast quality is not necessary correlated with its impact on management decisions. The paper fits the scope of the journal and has a clear potential for publication, given the increasing momentum of weather and climate services and how its "real" impact can be measured. I have no major concerns about the manuscript, although some improvements would further increase its quality. Therefore, I would consider it ready for publication after fixing the minor concerns I point at below.

We thank the referee for the positive comment.

TECHNICAL COMMENTS

1. Page 2, lines 28-31: In my opinion, the first sentence of this paragraph is just a summary of the previous one. I would delete it and reflect in the previous paragraph that an alternative promising metric would be the quality obtained on predicting decision- relevant variables.

   Following the reviewer comment, we will delete this sentence.

2. Page 3, line 23: Although it becomes clear when moving forward that "post-processing" means "downscaling and bias-correction", I would add a remark here just to clarify it.

   We thank the reviewer for the suggestion and we will clarify from the beginning the meaning of post-processing.

3. Page 4, line 6: what do you mean when you state "pilot"? I think it is a synonym of "case study", but sometimes the term "pilot" implies you run field experiments

to apply the method developed. Please clarify the term.

We agree with the reviewer that the term pilot might be misinterpreted. Since we used it a synonym for case study, we will remove this term in the revised version.

4. Page 5, lines 33-34: can you provide information to support the assumption of using crop yield as main driver of the cropping pattern decisions? Sometimes other variables like management complexity or profit predictability is more important than crop yield. In my opinion, you should clarify, if it is the case, that you make this assumption in the absence of more detailed information about the farmers' decision-making process.

We agree with the reviewer that this assumption should be clarified. In the revised version of the paper, we will mention that, in the absence of more detailed information about the farmers' decision-making process, we introduced this assumption on the basis of other similar studies (Hansen, 2004; Baigorria et al., 2008). *Hansen, J. (2004). "Linking dynamic seasonal climate forecasts with crop simulation for maize yield prediction in semi-arid Kenya". In: Agricultural and Forest Meteorology 125.1-2, pp. 143–157. Baigorria, G. a., J. W. Jones, and J. J. O'Brien (2008). "Potential predictability of crop yield using an ensemble climate forecast by a regional circulation model". In: Agricultural and Forest Meteorology 148.8-9, pp. 1353–1361.*

5. Page 6, line 21: as far as I know, the quantile-based mapping is a bias correction procedure. It is true that it has some downscaling component due to matching CDFs obtained at different spatial scales but, on a broader view, it is considered as a bias correction technique. In fact, you previously named it as a bias correction technique. Please fix this

Following the reviewer suggestion, we will fix this point in the revised version by consistently characterizing the quantile mapping as a bias-correction technique.

6. Page 8, lines 15-16: the way in which the aggregation is performed it is not clear. I assume you aggregate the daily data of the same month, but it may also mean you aggregate the ensemble members. Please clarify it. If you aggregate the ensemble members to obtain a unique factor, I would rather suggest keeping the factor obtained by each ensemble member and generate synthetic daily time series with all of them. In this way, you will have a better representation of the extremes, which are flattened when taking the average.

We agree with the reviewer that this step is not clear. We perform the following aggregations: first we aggregate the daily data of the same month, then we estimated a monthly perturbing factor for each ensemble members, and then we took the average factor across the ensemble's members. We are aware that in this way we lose some information on the extremes and we agree with the reviewer that performing the entire assessment on each single ensemble member would allow a better characterization of the extremes as well as exploring how this uncertainty is propagated when moving from the forecast quality to the operational value. Yet, this would be computationally challenging as it would require running 96 simulations per year, for a total of around 500 computational hours. This computational effort goes beyond the scope of this paper. Moreover, the use of large ensembles opens up a number of challenges (see the reply to the second point raised by R2) and the consequences of aggregating or not aggregating the ensemble members can be analyzed in detail, potentially focusing on a single forecast product, in a future work. In the revised version of the paper, we will clarify how we perform this aggregation and we will include this aspect in the list of assumption added in the conclusions section (see R1 suggestion), suggesting as a possible follow-up work the opportunity of refining our analysis keeping all the ensemble members separated.

7. Page 12, table 1: please include the ensemble members of each WCS used unless all the products provide just one ensemble. In this last case, you should

indicate in the text that all of them provide a unique ensemble member.

All the forecast products are in the form of ensembles: ECMWF products have 9 ensemble members (or 3 in case of decadal products), CanSIps have 10 ensemble members, CSFv2 has 4 ensemble members. We will include this information in the revised table 1.

8. Page 15, figure 6: from my point of view, the understandability of this figure would be increased by including the legend inside each individual plot as well as the name of the WCS product. Otherwise the reader needs to constantly go up and down the figure to find out what each bar refers to.

We agree with the reviewer comment and in the revised version we will move the legends inside the subplots as suggested to improve the interpretability of the figure.

9. Page 16, figure 7: same comment as for figure 6.

We agree with the reviewer comment and in the revised version we will move the legends inside the subplots as suggested to improve the interpretability of the figure.

10. Page 18, figure 8: I would include the name of the WCS product in each individual plot. Furthermore, I would also provide the value of an average score for the time series inside each plot (for example the MAE). In this way, the reader has a numerical way to easily compare the accuracy of each WCS product type for each plot.

We thank the reviewer for the suggestion and, in the revised version of the paper, we will move the legends of the subplots as suggested and we will add a numerical score to facilitate the comparison across WCS products.

11. Page 19, lines 1-5: Did you generate 100 time series for each year between 2001 and 2005? Did you choose one year between 2001 and 2005 and then generate

100 series for it? Or did you spare the 100 time series between 2001 and 2005? Please add a clarification about it.

We actually generated 100 time series for each year over the evaluation horizon (2001-2005). We will clarify this point in the revised version of the paper.

12. Page 20, lines 1-14: In my opinion, the fact that the neutral or optimistic risk profiles did not obtain the best performance for the best forecast deserves more explanation. How can you justify this issue? In the absence of more information, I would doubt about the suitability of the score used (median and variance of MAE). Maybe the IFS/HOPE product does not predict extremes as ECHAM5/MPIOM does, and due to this reason the latter offers the best performance on both the neutral and the optimistic risk profiles. Please add some explanation or theory about this unexpected finding.

This unexpected finding can be explained by the fact that forecast accuracy metrics quantify the error in predicting the agricultural production, while the operational value estimated through the decision model relies on the ranking of the available options (cropping patterns). Sub-optimal decisions are made when the forecasted productivity of the crops produces a different ranking with respect to the one resulting at the end of the agricultural season. However, such rank reversals are not linearly related to the forecast accuracy: large but consistent (e.g., systematic over/underestimation) errors for all the crops may produce the same ranking and result in optimal decisions, while smaller and variable errors can produce sub-optimal decisions. This is quite clear if we consider the forecast accuracy reported in Fig. 8 of ECMWF(annual) IFS/HOPE and ECHAM5/MPIOM: looking at the values in 2001, ECHAM5/MPIOM (which in Fig. 10 has the best performance) is systematically overestimating the productivity of all the crops; IFS/HOPE instead underestimates the productivity of tomato while overestimates the one of rice, potentially reverting the ranking of these crops and producing suboptimal decisions. Following the reviewer suggestion, we will clarify this point in

the revised manuscript.

---

## Referee Comment (RC3) · Anonymous Referee #3 · 20 Jul 2017

General comments: The paper is interesting and novel and it certainly falls within the scope of HESS. The paper presents a novel approach to evaluate climate predictions through the impacts they have on the user decisions. This is an important aspect in the evaluation of the predictions which is often overlook in the context of climate services. The paper try to reach some substantial and interesting conclusions but the results are somehow weakened by the design of the experiments and the methodology that has been followed. The assumptions made are clearly outlined but the scientific methods (bias-correction) and datasets used (ENSEMBLES) lag a bit behind what I would consider the current state of the art.

[Figure]

Specific comments: More information on the bias correction methodology should be provided to allow the reproduction of the results by fellow scientists. In particular reading section 3 it is not clear whether the bias correction is applied to the forecast on a lead-time basis or weather instead the author perform the Q-Q bias correction using a CDF obtained looking at the entire forecast period. If, as it seems, it is the latter, the approach is likely to lead to incorrect results as the forecast bias is lead-time dependent (e.g. Doblas-Reyes et al 2013) whilst the CDF would be calculated on a full 7 month forecast. This is unlikely to be a major problem in regions characterise by a limited seasonal cycle and a small model drift as you could assume the relationship linking model output and observations to be roughly the same throughout the year. Unfortunately I don't think such an assumption would hold in the region of study.

The paper appears to be based on a set of seasonal prediction ensembles characterised by a relatively small ensemble size. Given that we now know that, at least in the case of the NAO in Europe, the climate model signal strength depends on the number of ensemble members (e.g. Scaife et al. 2014) the results presented here may significantly under represent the real usefulness of seasonal climate prediction for the target users.

As noted by other reviewers the evaluation was made on an extremely short time period something which can only further reduce the significance of the results.

In the light of the points raised above I am not convinced the approach, despite its novelty and user-consideration, is necessarily fair in the analysis of the seasonal predictions and their value for informing decision makers.

Technical comments:

Weather and Climate Services (WCS) is not an acronym I came across before. Given the fundamental difference between the way in which climate and weather model output are typically dealt with I am not sure this is particularly useful. Furthermore World Climate Services. (WCS) is also a trade name of a MeteoGroup product.
Stream 2 was an experiment in the context of ENSEMBLE project rather than a project per-se as erroneously stated in section 4.

The statement about usefulness of seasonal prediction in agricultural application that appears in line 9 of the abstract is too general too be correct as there are regions of the world where these kind of predictions are known to be usable and useful.

Cloke and Pappenberger 2009 doesn't strike as being the most relevant reference to describe the recent development of WCS especially considering is nearly 10 years old now.

---

## Author Response (AR1)

**POLITECNICO**
MILANO 1863

Reply to reviewers about paper HESS-2017-304

**A coupled human-natural system to assess the operational value of weather and climate services for agriculture**

Y. Li, M. Giuliani, A. Castelletti

Dear Editor,

We would like to thank you and the three reviewers for the thorough and very helpful review.

We took all your points into consideration and revised the manuscript accordingly.

A detailed reply to reviewers is attached below. In preparing our response, all references to line numbers, equations, and figures are based on the revised manuscript; authors' replies as well as the changes tracked in the manuscript are in blue.

Sincerely,

[Figure]

**Editor**

I agree with the reviewers that this paper deals with an important area of research and fits in with this special issue well. The three reviews were very thorough. The authors should address their comments comprehensively. In particular, the limitation of using a very short evaluation period needs to be emphasised when drawing conclusions from this study, as I do not believe it is possible to reach any robust conclusions from such a short period when evaluating seasonal forecasts and their applications. An additional comment from me is that the authors did consider irrigation water supply and management even though the title of the paper is about irrigated agriculture. In fact, there is a lot more skill in forecasting streamflow, which is related to water availability, than climate. The description in the paper gave me the impression that it was about rain fed agriculture.

We thank the editor for the positive comment and we agree on the very helpful comments provided by the three reviewers. We took all the points in consideration and revised the manuscript accordingly.

In particular, we added a discussion about the main assumptions and limitations of our work in the conclusion section (page 22, lines 10-30), where we also clarified the motivation for limiting the analysis to the time period (2001-2005): 1) the historical observations available for running the model covers the period (1993-2005), which were divided into two periods with the first period used for post-processing the forecast products and the second one for performing the analysis; 2) ECMWF forecast products are obtained from the "ENSEMBLES" project, which provides hindcasts over the period (1960-2005); 3) CSF v2 and CanSIps cover the period (1981-2010), but they are outperformed by ECMWF products.

As for the comment about the use of "irrigated agriculture" in the title, we agree that this can be misinterpreted. We mentioned irrigated agriculture as the considered study site is an irrigation district and not a rain-fed agricultural area. However, the irrigation supply to the agricultural districts is regulated since 1946 through the operations of Lake Como. This ensures a reliable irrigation supply to the farmers during the agricultural season, who hence become interested in predictions of future climate. Moreover, the regulation of the lake represents a challenge for the hydrological models used for seasonal streamflow predictions, ultimately reducing their forecast skills. In any case, to avoid confusion we changed the title in "A coupled human-natural system to assess the operational value of weather and climate services for agriculture".

**Referee comment #1**

The subject of the paper "*A coupled human-natural system to assess the operational value of weather and climate services for irrigated agriculture*" is of direct interest to the Journal of Hydrology and Earth System Sciences. Authors introduce and apply a framework in the context of measuring the operational value of weather and climate services (WCs). The validation of the usefulness of the WCs to the final users is a much needed step towards the realization of these services.

SPECIFIC COMMENTS
1.  One of my concerns is the limited duration of the analysis period (2001-2005). Why authors didn't extend the analysis beyond 2005. Is it due to the limited data availability? If yes, it would be also interesting to see similar results for a longer time period even for less forecast products.
The motivation for limiting the analysis to the time period (2001-2005) is manifold: 1) the historical observations available for running the model covers the period (1993-2005), which were divided into two periods with the first period used for post-processing the forecast products and the second one for performing the analysis; 2) ECMWF forecast products are obtained from the "ENSEMBLES" project, which provides hindcasts over the period (1960-2005); 3) CSF v2 and CanSIps cover the period (1981-2010), but they are outperformed by ECMWF products.
We clarified this point in the discussion of limitations/assumptions of the study that we included in the conclusion section (see page 22, lines 10-30).

2.  Assuming that instead a single forecast product, a large ensemble developed by the combination of several products could outperform the forecast quality or the result to better decisions compared to single products?
We agree with the reviewer that a larger ensemble (note that all the products we used are in the form of ensemble forecast) might attain a better performance in terms of forecast quality and, possibly, also in terms of operational value. However, the use of multi-model ensembles opens up a number of challenges - such as how to limit the smoothing effect on the extreme events, how to combine multiple products with different levels of accuracy, how to simplify the uptake of the resulting large ensemble - which goes beyond the scope of this paper and can be explored in a future analysis. In the revised version of the paper, we included this point in the list of limitations/assumptions added to the conclusions section, suggesting the opportunity of exploring it in a future research (see page 22, lines 10-30).

3.  Another subject that could also be discussed is the limitations and/or assumptions of the study. I think that a limitations section should be added in the paper in order to summarize the main simplifications or assumptions considered in the work. For example the determinant yield factor is the water availability no matter the agricultural treating of the farmers during the cultivation period. Maybe such a section could also include some references to works in which they have been treated in other way.
The point raised by the reviewer is well taken. In this work, we didn't explore the impacts of agricultural practices (primarily the use of nutrients and fertilizers) as the water availability is predominant in the considered case study. The validity of this assumption is discussed in our previous work (Giuliani et al., 2016). However, we agree that other determinants factors can be explored in a future work. Following the reviewer suggestion, we added a list of

limitations/assumptions of the study in the conclusion section of the revised manuscript. This list (see page 22, lines 10-30) includes the limited duration of the analysis (see point 1), the limited exploration of the socio-economic dimension of the problem (i.e. the prediction of crops' prices which are instead assumed as deterministic in the current analysis), the assumption that crop yield determined by water availability is the main driver of the cropping pattern decisions.

*Giuliani, M., Li, Y., Castelletti, A., and Gandolfi, C.: A coupled human-natural systems analysis of irrigated agriculture under changing climate, Water Resources Research, 52, 6928–6947, doi:10.1002/2016WR019363, 2016.*

Considering these and the fact that the scientific significance and quality are excellent, my suggestion to the editors would be to accept after minor revision in the context of my specific and technical comments. I am listing a number of suggestions in the form of technical comments that will improve the presentation of the study.
We thank the referee for the positive comment and for his/her thorough review of the paper which will contribute in improving the presentation of the study.

TECHNICAL COMMENTS
P4 – Study site section: since you are dealing with end-user services it would be nice if you include more information, a short description of the users (total number of farmers, average farm extent, etc.).
Following the reviewer suggestion, in the revised version we included additional information on the case study and the users (e.g., the Muzza irrigation district is characterized by a hierarchical structure, which includes 39 irrigation units at the first level, which can be further partitioned in 1722 "comizi" at the second level - see page 4 lines 8-10).

P4 – L9: Here you mention 40% for maize while in Fig. 1 shows 74%.
We thank the reviewer for noticing this discrepancy, which is due to a typo in the text, while the legend is correct. We will fix it in the revised version of the paper (page 4 line 7).

P5 – L2: you could also say that climate change has exacerbated the severity of the extreme events (drought/heat wave).
We agree with the reviewer suggestion and we will modify this sentence accordingly (page 4 line 12).

P5 – L5: what about 2001? Judging from Fig. 4 2001 was even drier than 2003 and 2005 (also in Fig. 6 for the April to August precipitation).
This figure is probably not able to fully characterize the variability across the considered year. In fact the reviewer is correct in saying that 2001 seems drier than 2003 and 2005. However, in 2001 there were abundant precipitations in winter which allowed storing water in the form of snow and in the Lake Como, thus ensuring adequate irrigation during the agricultural season. Conversely, 2003 and 2005 had also a dry winter, thus facing the most critical conditions for the agricultural activities in the basin. Following also another comment on this figure, in the revised manuscript we added information about the total annual precipitation and average temperature to allow distinguishing the conditions in 2001 with respect to the ones in 2003 and 2005.

P5 – L26: add cross reference for Table 1.

Following the reviewer suggestion, we will add a cross-reference to the table in the revised manuscript (page 5 line 5).

P9 – L5: Why do you set the resolution to 250m? Is this resolution adequate for representing the spatial detail of the crops/properties?

The resolution of the model was set in previous works (e.g., Vassena el al., 2012) to allow a proper characterization of the spatial distributions of all the components of the model, especially in terms of water balance module. We clarified this point in the revised manuscript (page 9 lines 7-9).

*Vassena et al. (2012), Modeling water resources of a highly irrigated alluvial plain (Italy): calibrating soil and groundwater models, Hydrogeology journal, 20(3): 449-467.*

P9 – L25: Does this model take into account the behavioral dependency on the preceding year? Meaning that the farmers' decision is affected for example from a "previous (i-1) dry year" and as a result the potentially optimistic decision of year i would be more pessimistic?

In principle, our model can account for this type of behavioral dependency. However, the calibration of a decision model implementing such behavioral dependency requires long behavioral time series to identify the proper lag-time as well as the magnitude of the effect for different levels of drought intensity. In the absence of such large observational dataset, we decided to partially explore this point by 1) simulating farmers' decisions made assuming the next year is equal to the previous one or to the average of the last two (see EmpPast and Emp2Ave experiments); 2) running a sensitivity analysis using different levels of risk aversion. In the revised version of the paper, we included this point in the list of limitations added to the conclusions section, suggesting the opportunity of exploring it in a future research where enough observational data can be available (see page 22, lines 10-30).

P12 – Table1: the products listed here are single member experiments or there is a number of realizations?

All the forecast products are in the form of ensembles: ECMWF products have 9 ensemble members (or 3 in case of decadal products), CanSIps have 10 ensemble members, CSFv2 has 4 ensemble members. We included this information in the revised table 1.

P13 – Figure 4: You could also add total precipitation and average temperature for each year (row) on the right part of the figure (on the left from the legend).

We thank the reviewer for this useful suggestion (which also allow solving the other issue of Figure 4). In the revised version of the paper, we added information about the total annual precipitation and average temperature of each year.

P15 – Figure 6: it would be easier to read if you place the legend of each product on the corresponding sub- plot. Otherwise you could arrange the legend in similar order as the subplots because it is hard to detect. It would be also helpful if you could highlight the dry years.

Following the reviewer suggestion (shared also by R#2), in the revised version we moved the legends inside the subplots as suggested to improve the interpretability of the figure.

P16 – Figure 7: The differences are hard to distinguish. You could plot the anomalies instead or adjust the range of the temperature axis (for example from 17 to 23oC). Again it would be also helpful if you could highlight the dry years.

We agree with the reviewer and, following his/her suggestion we modified the figure by adjusting the range of the temperature axis and by moving the legends inside the subplots.

P18 – Figure 8: You could use a continuous line for the deterministic simulation.

We thank the reviewer for the suggestion and, in the revised version of the paper, we modified the figure accordingly.

P23 – Line 32: remove the space from "f armers"

We fixed the typo in the revised version (page 23 line 24).

**Referee comment #2**

The presented manuscript describes and applies a methodological framework to assess the operational value of weather and climate forecast products on irrigated agriculture. It combines a set of forecast products with an agronomic model that simulates the crop yield based on meteorological inputs and an agent-based model that establishes the optimal cropping pattern depending on the forecasts available and the risk profile of the farmers. The novelty of the paper consists in the joint assessment of the forecast quality and its impact on management decisions and farmers risk profile. The methodology is well described and the structure and organization of the paper is coherent and adequate. The results point at the fact that the forecast quality is not necessary correlated with its impact on management decisions.

The paper fits the scope of the journal and has a clear potential for publication, given the increasing momentum of weather and climate services and how its "real" impact can be measured. I have no major concerns about the manuscript, although some improvements would further increase its quality. Therefore, I would consider it ready for publication after fixing the minor concerns I point at below.

We thank the referee for the positive comment.

TECHNICAL COMMENTS

1. Page 2, lines 28-31: In my opinion, the first sentence of this paragraph is just a summary of the previous one. I would delete it and reflect in the previous paragraph that an alternative promising metric would be the quality obtained on predicting decision- relevant variables.

Following the reviewer comment, we deleted this sentence.

2. Page 3, line 23: Although it becomes clear when moving forward that "post- processing" means "downscaling and bias-correction", I would add a remark here just to clarify it.

We thank the reviewer for the suggestion and we clarified from the beginning the meaning of post-processing (page 3 line 22).

3. Page 4, line 6: what do you mean when you state "pilot"? I think it is a synonym of "case study", but sometimes the term "pilot" implies you run field experiments to apply the method developed. Please clarify the term.

We agree with the reviewer that the term pilot might be misinterpreted. Since we used it a synonym for case study, we removed this term in the revised version.

4. Page 5, lines 33-34: can you provide information to support the assumption of using crop yield as main driver of the cropping pattern decisions? Sometimes other variables like management complexity or profit predictability is more important than crop yield. In my opinion, you should clarify, if it is the case, that you make this assumption in the absence of more detailed information about the farmers' decision-making process.

We agree with the reviewer that this assumption should be clarified. In the revised version of the paper (page 5 lines 10-11), we mentioned that, in the absence of more detailed information about the farmers' decision-making process, we introduced this assumption on the basis of other similar studies (Hansen, 2004; Baigorria et al., 2008).

*Hansen, J. (2004). "Linking dynamic seasonal climate forecasts with crop simulation for maize yield prediction in semi-arid Kenya". In: Agricultural and Forest Meteorology 125.1-2, pp. 143–157.*
*Baigorria, G. a., J. W. Jones, and J. J. O'Brien (2008). "Potential predictability of crop yield using an ensemble climate forecast by a regional circulation model". In: Agricultural and Forest Meteorology 148.8-9, pp. 1353–1361.*

5. Page 6, line 21: as far as I know, the quantile-based mapping is a bias correction procedure. It is true that it has some downscaling component due to matching CDFs obtained at different spatial scales but, on a broader view, it is considered as a bias correction technique. In fact, you previously named it as a bias correction technique. Please fix this

Following the reviewer suggestion, we fixed this point by consistently characterizing the quantile mapping as a bias-correction technique (page 6 line 24).

6. Page 8, lines 15-16: the way in which the aggregation is performed it is not clear. I assume you aggregate the daily data of the same month, but it may also mean you aggregate the ensemble members. Please clarify it.

If you aggregate the ensemble members to obtain a unique factor, I would rather suggest keeping the factor obtained by each ensemble member and generate synthetic daily time series with all of them. In this way, you will have a better representation of the extremes, which are flattened when taking the average.

We agree with the reviewer that this step is not clear. We perform the following aggregations: first we aggregate the daily data of the same month, then we estimated a monthly perturbing factor for each ensemble members, and then we took the average factor across the ensemble's members. We are aware that in this way we lose some information on the extremes and we agree with the reviewer that performing the entire assessment on each single ensemble member would allow a better characterization of the extremes as well as exploring how this uncertainty is propagated when moving from the forecast quality to the operational value. Yet, this would be computationally challenging as it would require running 96 simulations per year, for a total of around 500 computational hours. This computational effort goes beyond the scope of this paper. Moreover, the use of large ensembles opens up a number of challenges (see the reply to the second point raised by R#2) and the consequences of aggregating or not aggregating the ensemble members can be analyzed in detail, potentially focusing on a single forecast product, in a future work. In the revised version of the paper, we clarified how we perform this aggregation (page 8 lines 17-19) and we included this aspect in the list of assumptions added in the conclusions section (see R#1 suggestion), suggesting as a possible follow-up work the opportunity of refining our analysis keeping all the ensemble members separated (see page 22, lines 10-30).

7. Page 12, table 1: please include the ensemble members of each WCS used unless all the products provide just one ensemble. In this last case, you should indicate in the text that all of them provide a unique ensemble member.

All the forecast products are in the form of ensembles: ECMWF products have 9 ensemble members (or 3 in case of decadal products), CanSIps have 10 ensemble members, CSFv2 has 4 ensemble members. We added this information in the revised table 1.

8. Page 15, figure 6: from my point of view, the understandability of this figure would be increased by including the legend inside each individual plot as well as the name of the WCS

product. Otherwise the reader needs to constantly go up and down the figure to find out what each bar refers to.

We agree with the reviewer comment and, as suggested, in the revised version we moved the legends inside the subplots.

9. Page 16, figure 7: same comment as for figure 6.

We agree with the reviewer comment and in the revised version we moved the legends inside the subplots.

10. Page 18, figure 8: I would include the name of the WCS product in each individual plot. Furthermore, I would also provide the value of an average score for the time series inside each plot (for example the MAE). In this way, the reader has a numerical way to easily compare the accuracy of each WCS product type for each plot.

We thank the reviewer for the suggestion and, in the revised version of the paper, we moved the legends of the subplots. As suggested, we also added the mean average error of each product in predicting the crop productivity.

11. Page 19, lines 1-5: Did you generate 100 time series for each year between 2001 and 2005? Did you choose one year between 2001 and 2005 and then generate 100 series for it? Or did you spare the 100 time series between 2001 and 2005? Please add a clarification about it.

We actually generated 100 time series for each year over the evaluation horizon (2001-2005). We clarified this point in the revised version of the paper (page 19 lines 5-6).

12. Page 20, lines 1-14: In my opinion, the fact that the neutral or optimistic risk profiles did not obtain the best performance for the best forecast deserves more explanation. How can you justify this issue? In the absence of more information, I would doubt about the suitability of the score used (median and variance of MAE). Maybe the IFS/HOPE product does not predict extremes as ECHAM5/MPIOM does, and due to this reason the latter offers the best performance on both the neutral and the optimistic risk profiles. Please add some explanation or theory about this unexpected finding.

This unexpected finding can be explained by the fact that forecast accuracy metrics quantify the error in predicting the agricultural production, while the operational value estimated through the decision model relies on the ranking of the available options (cropping patterns). Sub-optimal decisions are made when the forecasted productivity of the crops produces a different ranking with respect to the one resulting at the end of the agricultural season. However, such rank reversals are not linearly related to the forecast accuracy: large but consistent (e.g., systematic over/underestimation) errors for all the crops may produce the same ranking and result in optimal decisions, while smaller and variable errors can produce sub-optimal decisions. This is quite clear if we consider the forecast accuracy reported in Fig. 8 of ECMWF(annual) IFS/HOPE and ECHAM5/MPIOM: looking at the values in 2001, ECHAM5/MPIOM (which in Fig. 10 has the best performance) is systematically overestimating the productivity of all the crops; IFS/HOPE instead underestimates the productivity of tomato while overestimates the one of rice, potentially reverting the ranking of these crops and producing sub-optimal decisions. Following the reviewer suggestion, we clarified this point in the revised manuscript (page 20 lines 11-14; page 21 lines 1-6).

**Referee comment #3**

General comments: The paper is interesting and novel and it certainly falls within the scope of HESS. The paper presents a novel approach to evaluate climate predictions through the impacts they have on the user decisions. This is an important aspect in the evaluation of the predictions which is often overlook in the context of climate services. The paper try to reach some substantial and interesting conclusions but the results are somehow weakened by the design of the experiments and the methodology that has been followed. The assumptions made are clearly outlined but the scientific methods (bias-correction) and datasets used (ENSEMBLES) lag a bit behind what I would consider the current state of the art.

Specific comments: More information on the bias correction methodology should be provided to allow the reproduction of the results by fellow scientists. In particular reading section 3 it is not clear whether the bias correction is applied to the forecast on a lead-time basis or weather instead the author perform the Q-Q bias correction using a CDF obtained looking at the entire forecast period. If, as it seems, it is the latter, the approach is likely to lead to incorrect results as the forecast bias is lead-time dependent (e.g. Doblas-Reyes et al 2013) whilst the CDF would be calculated on a full 7 month forecast. This is unlikely to be a major problem in regions characterise by a limited seasonal cycle and a small model drift as you could assume the relationship linking model output and observations to be roughly the same throughout the year. Unfortunately I don't think such an assumption would hold in the region of study.

We agree that part of Section 3 was probably not completely clear. Specifically, given the strong intra-annual seasonal cycle of our study site, the bias-correction was applied on a monthly basis and not using a CDF calculated on the full 7 month forecast period. We clarified this point in the revised manuscript (page 6 lines 20-23).

The paper appears to be based on a set of seasonal prediction ensembles characterised by a relatively small ensemble size. Given that we now know that, at least in the case of the NAO in Europe, the climate model signal strength depends on the number of ensemble members (e.g. Scaife et al. 2014) the results presented here may significantly under represent the real usefulness of seasonal climate prediction for the target users.

We agree with the reviewer comment – which is shared by other reviewers - that a larger ensemble (note that all the products we used are in the form of forecasts' ensemble) might attain a better performance in terms of forecast quality and, possibly, also in terms of operational value. However, the use of large ensembles, potentially multi-model ensembles, opens up a number of challenges - such as how to limit the smoothing effect on the extreme events, how to combine multiple products with different levels of accuracy, how to simplify the uptake of the resulting large ensemble - which goes beyond the scope of this paper and can be explored in a future analysis. We clarified this point in the discussion of limitations of the study that we added in the conclusion section as suggested by R#1 (see page 22 lines 10-30).

As noted by other reviewers the evaluation was made on an extremely short time period something which can only further reduce the significance of the results.

In the light of the points raised above I am not convinced the approach, despite its
novelty and user-consideration, is necessarily fair in the analysis of the seasonal predictions
and their value for informing decision makers.

The motivation for limiting the analysis to the time period (2001-2005) is manifold: 1) the historical
observations available for running the model covers the period (1993-2005), and we used the first
period for post-processing the forecast products and the second one for performing the analysis; 2)
ECMWF forecast products are obtained from the "Ensemble" project, which provides hindcasts
over the period (1960-2005); 3) CSF v2 and CanSIps cover the period (1981-2010), but they are
outperformed by ECMWF products.

We clarified this point in the discussion of limitations/assumptions of the study that we added in the
conclusion section as suggested by R#1 (see page 22 lines 10-30).

Technical comments:
Weather and Climate Services (WCS) is not an acronym I came across before. Given
the fundamental difference between the way in which climate and weather model output
are typically dealt with I am not sure this is particularly useful. Furthermore World
Climate Services. (WCS) is also a trade name of a MeteoGroup product.

By googling WCS we found the acronym with the meaning it was used in the paper. In any case, to
avoid confusion, we changed into W&C Services everywhere across the paper.

Stream 2 was an experiment in the context of ENSEMBLE project rather than a project
per-se as erroneously stated in section 4.

We thank the reviewer for pointing this out. In the revised version of the manuscript we specified
that Stream2 was part of ENSEMBLE project (page 12 line 4).

The statement about usefulness of seasonal prediction in agricultural application that
appears in line 9 of the abstract is too general too be correct as there are regions of
the world where these kind of predictions are known to be usable and useful.

We agree with the reviewer that this sentence a too vague. In the revised version of the manuscript,
we modified it by specifying that this conclusion holds for the case study analyzed in the paper
(page 1 lines 8-11).

Cloke and Pappenberger 2009 doesn't strike as being the most relevant reference to
describe the recent development of WCS especially considering is nearly 10 years old
now.

Following the reviewer suggestion of citing more recent works, in the revised version of the paper
we added the following references (page 1, line 23-24): 1) *Bauer, Peter, Alan Thorpe, and Gilbert

[revised manuscript text omitted]